# MolSpectLLM: A Molecular Foundation Model Bridging Spectroscopy, Molecule Elucidation, and 3D Structure Generation

## Abstract

Recent advances in molecular foundation models have shown impressive performance in molecular property prediction and *de novo* molecular design, with promising applications in areas such as drug discovery and reaction prediction. Nevertheless, most existing approaches rely exclusively on SMILES representations and overlook both experimental spectra and 3D structural information—two indispensable sources for capturing molecular behavior in real-world scenarios. This limitation reduces their effectiveness in tasks where stereochemistry, spatial conformation, and experimental validation are critical. To overcome these challenges, we propose **MolSpectLLM**, a molecular foundation model pretrained on Qwen2.5-7B that unifies experimental spectroscopy with molecular 3D structure. By explicitly modeling molecular spectra, MolSpectLLM achieves state-of-the-art performance on spectrum-related tasks, with an average accuracy of 0.53 across NMR, IR, and MS benchmarks. MolSpectLLM also shows strong performance on the spectra analysis task, obtaining 15.5% sequence accuracy and 41.7% token accuracy on Spectra-to-SMILES, substantially outperforming large general-purpose LLMs. More importantly, MolSpectLLM not only achieves strong performance on molecular elucidation tasks, but also generates accurate 3D molecular structures directly from SMILES or spectral inputs, bridging spectral analysis, molecular elucidation, and molecular design.

## 1 Introduction

In recent years, the rapid development of large language models (LLMs) has captured widespread attention across academia and industry (Brown et al., 2020; Devlin et al., 2019; Achiam et al., 2023). Building on these advances, researchers have extended the foundation model paradigm beyond natural language, adapting large-scale architectures and training strategies to the molecular sciences. This emerging class of molecular foundation models leverages vast chemical datasets to enable knowledge transfer across diverse tasks in chemistry, biology, and drug discovery (Liu et al., 2023; Wang et al., 2024; García-Ferrero et al., 2024; Zhang et al., 2024; Zhao et al., 2025; Xia et al., 2025; Liu et al., 2024; Tan et al., 2025).

Recent efforts have demonstrated strong performance in molecular property prediction (Tan et al., 2025), reaction outcome forecasting (Tharwani et al., 2025; Shi et al., 2023), and *de novo* molecular design (Tan et al., 2025; Jiang et al., 2025), underscoring the transformative potential of this line of research. However, most existing molecular foundation models rely predominantly on simplified string-based representations such as SMILES (Weininger, 1988). While compact and convenient for large-scale pretraining, SMILES inherently discards two critical sources of information: (i) three-dimensional (3D) molecular structure, which governs stereochemistry, conformational dynamics, and intermolecular interactions (Greer et al., 1994; Schuur et al., 1996); and (ii) experimental molecular spectra, which provide rich empirical signals of molecular identity and composition through techniques such as nuclear magnetic resonance (NMR) (Keeler, 2011), infrared (IR) (Bellamy, 2013), and mass spectrometry (MS) (McLafferty, 1993). Neglecting these modalities limits the capacity of current models to reason over real-world molecular behavior, and restricts their applicability to tasks where experimental validation and 3D structural fidelity are essential.

To address these gaps, we propose **MolSpectLLM**, a molecular foundation model that integrates three complementary modalities: molecular spectroscopy, molecular elucidation, and 3D structural reasoning. By explicitly modeling NMR, IR, and MS spectra, MolSpectLLM demonstrates strong capabilities in spectrum-related tasks, including spectral interpretation and molecular identification. Beyond spectroscopy, MolSpectLLM achieves competitive or state-of-the-art performance on diverse molecular benchmarks, such as molecule QA, SMILES–IUPAC name conversions, and property prediction. Most importantly, MolSpectLLM is capable of generating accurate three-dimensional molecular structures directly from textual or spectral inputs, thereby bridging empirical measurement with structural understanding and generative molecular design. We argue that such integration is a crucial step toward developing versatile and reliable molecular foundation model, advancing applications in chemistry, biology, and materials science.

The key contributions of this work are summarized as follows:

- We incorporate molecular spectroscopy and 3D structural information into a unified molecular foundation model, achieving strong performance across diverse tasks.

- We design a series of standard textual description tailored to the characteristics of different spectral modalities (e.g., NMR, IR, and MS). These representations efficiently encode spectral information, enabling large language models to effectively interpret and leverage experimental data.

- We propose new evaluation metrics for spectrum generation that operate directly on the generated textual descriptions, eliminating the need to convert them back into sparse raw vectors. This design provides a more direct, interpretable, and efficient means of assessing spectral fidelity.

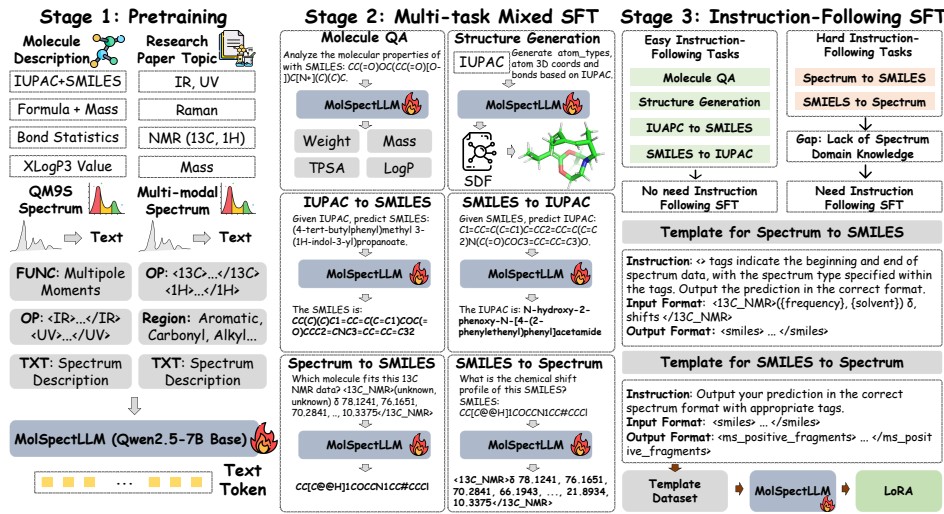

Figure 1: Pipeline of MolSpectLLM. The training of MolSpectLLM consists of three stages. During pretraining, we leverage publicly available chemical literature and unified textual descriptions constructed from PubChem, QM9S, and Multi-modal Spectrum. Then we perform instruction tuning on curated instruction datasets, followed by lightweight LoRA adaptation on a small set of template-based data to facilitate evaluation. Further details are provided in Section 3.5 and A.2.

## 2 RELATED WORK

**Molecular Foundation Models** Molecular foundation models have recently emerged as powerful tools for molecular property prediction and *de novo* molecular design. Early works such as ChemLLM (Zhang et al., 2024), ChemMLLM (Tan et al., 2025) and NatureLM (Xia et al., 2025) adapted language model architectures to text-based chemical representations (formula, SMILES, IUPAC Name), achieving strong results in molecular property benchmarks. However, the reliance on simplified representations neglects the 3D structural information and stereochemistry of molecules (Zhao et al., 2025; Mirza et al., 2025). In an attempt to address this gap, more recent advances, including GraphMVP (Liu et al., 2022) and GeoMol (Ganea et al., 2021), explicitly leverage 3D geometric information to learn richer molecular embeddings. Besides, most existing approaches remain limited to purely computational representations, often overlooking experimental data sources

such as spectroscopy, which is not generalizable to real-world chemical tasks (Rashed & Gorislav, 2024; Salimova et al., 2025; Luo et al., 2023). To tackle this limitation, our method integrates computational molecular representations and spectroscopic data into a unified framework, thereby enhancing its adaptability to practical chemical scenarios.

**Leveraging Spectroscopy in Molecular Modeling**  Molecular spectroscopy provides direct structural and compositional information of molecules (Yang et al., 2025; Elias et al., 2004; Prasad et al., 2025). This rich experimental data serves as a crucial bridge between computational models and real-world chemistry. However, most molecular foundation models have yet to incorporate experimental spectroscopy as an input modality, leaving a gap in bridging experimental evidence with predictive molecular modeling (Zhang et al., 2024; Tan et al., 2025). Among the few smaller-scale models that have attempted to leverage spectroscopy, most adopt a naive, end-to-end sequence-based approach (Litsa et al., 2023; Liu et al., 2017), which is limited by specific task types and spectral types. When applying tasks or spectral formats outside the scope of its training data, it often fails to maintain reliable performance. Unlike SpectraLLM (Su et al., 2025), which simply reformulates spectra into natural language and is limited to predicting SMILES, our approach introduces standardized textual descriptions of spectra and extends beyond structure elucidation to spectrum generation, 3D structure prediction, and broader molecular understanding tasks.

# 3 MOLSPECTLLM

## 3.1 OVERVIEW

We propose **MolSpectLLM**, a large-scale molecular foundation model bridging molecular spectroscopy with molecular elucidation and three-dimensional (3D) structure generation as illustrated in Fig. 1. In contrast to existing approaches that rely solely on SMILES representations, Mol-SpectLLM incorporates both 3D structural information and experimental spectra. To enable the language model to effectively interpret diverse and inherently sparse spectral vectors, we extract features tailored to the characteristics of each spectral modality and transform them into a standardized textual description.

## 3.2 SPECTRUM TEXTUAL DESCRIPTION

**Challenges in processing molecular spectra with LLMs.**  Molecular spectra encode essential experimental information for elucidating molecular structure. However, directly feeding raw spectral vectors into LLMs is ineffective: spectra are typically high-dimensional yet extremely sparse, with most entries containing no signal, and experimental spectra often contain substantial noise. These issues make it difficult for LLMs to extract chemically meaningful patterns, leading to poor performance in downstream reasoning tasks.

**Standardized textual representations.**  To make spectra more interpretable for LLMs, we design structured textual formats tailored to the characteristics of each modality (Fig. 2). For $^{13}$C NMR, we extract and serialize the chemical shift values (Fig. 2A). For $^{1}$H NMR, we additionally encode multiplicities, integration values, frequency, and solvent, yielding the representation shown in Fig. 2B. Here, `centroid` denotes the chemical shift (ppm), `shape` specifies multiplicity (e.g., singlet, doublet, triplet), `j_str` records coupling constants ($J$ values, Hz), and `nH` gives the number of protons derived from integration. For waveform-based spectra such as Raman, UV, and IR, we first apply interpolation smoothing and remove low-intensity noise peaks, then encode the cleaned spectra as value ranges and frequency–intensity pairs (Fig. 2C). For mass spectrometry (MS), we explicitly record the acquisition mode and collision energy in the tags, while each entry stores an $m/z$ value with its relative abundance (Fig. 2D).

By standardizing all spectra into compact textual forms, we remove sparsity and noise while retaining chemically meaningful features. This enables LLMs to interpret spectral information in a structured and consistent manner, forming the basis for accurate spectrum-to-structure and structure-to-spectrum modeling. Moreover, incorporating multiple spectrum modalities brings complementary benefits: IR highlights functional groups through characteristic bond vibrations (Stuart, 2004), NMR provides atomic-level resolution of connectivity and stereochemistry (Claridge, 2016), and

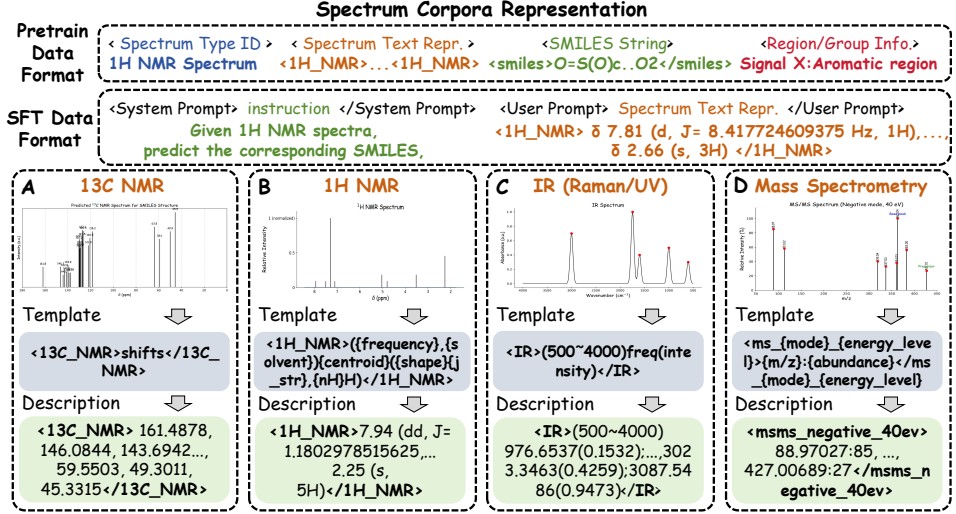

Figure 2: Standard textual description for different spectrum types. Instead of using raw spectral vectors, we design spectrum-specific feature extraction pipelines and convert the results into structured textual formats for LLM consumption. Details of the data processing are described in Sec. 3.2 and Appendix A.2.

MS reveals molecular weight and fragmentation signatures (Gross, 2017). Together, these modalities supply orthogonal constraints that guide the model toward chemically valid and structurally consistent predictions.

## 3.3 SPECTRUM ASSESSMENT

**(1) NMR Spectrum Generation.** We design a dedicated evaluation protocol for assessing $^{13}C$ and $^{1}H$ NMR generation. The key idea is to match predicted peaks against ground-truth peaks within specified ppm tolerances, and then aggregate peak-wise matches into standard set- and error-based scores. Below, we formalize the metric definitions and clarify the notation.

**(1.1) $^{13}C$ NMR (peak set without intensities).** Let the ground-truth carbon shifts be the multiset $C = \{\delta_i^{(C)}\}_{i=1}^{n_{\text{true}}}$ and the predictions $\widehat{C} = \{\widehat{\delta}_j^{(C)}\}_{j=1}^{n_{\text{pred}}}$, where each $\delta$ denotes a chemical shift in ppm. We construct a one-to-one matching $M \subseteq \{1, \ldots, n_{\text{pred}}\} \times \{1, \ldots, n_{\text{true}}\}$ using greedy nearest-neighbor assignment subject to a tolerance $\tau_C = 0.5$ ppm:

$$(j, i) \in M \iff \left|\widehat{\delta}_j^{(C)} - \delta_i^{(C)}\right| \leq \tau_C$$

and $i$ is unmatched, selecting the unused $i$ with minimal absolute difference.

Let $n_{\text{match}} = |M|$ be the number of matched pairs, with per-match deviations $d_{(j,i)} = \left|\widehat{\delta}_j^{(C)} - \delta_i^{(C)}\right|$. We report (per spectrum and averaged across the dataset):

$$P = \frac{n_{\text{match}}}{n_{\text{pred}}}, \quad R = \frac{n_{\text{match}}}{n_{\text{true}}}, \quad \text{F1} = \frac{2PR}{P+R}, \quad \text{MAE} = \frac{1}{n_{\text{match}}} \sum_{(j,i) \in M} d_{(j,i)}.$$

**(1.2) $^{1}H$ NMR (peaks with integration).** Each proton peak is represented as a tuple (shift, integration), where integration denotes the number of equivalent protons. The ground-truth list is $H = \{(\delta_i^{(H)}, nH_i^{(\text{true})})\}_{i=1}^{N_{\text{true}}}$, and the predictions are $\widehat{H} = \{(\widehat{\delta}_j^{(H)}, nH_j^{(\text{pred})})\}_{j=1}^{N_{\text{pred}}}$. We build a one-to-one *weighted* matching $\widehat{M}$ by scanning each prediction and assigning it to the unused ground-truth peak within a tolerance $\tau_H = 0.12$ ppm that *maximizes* a Gaussian-decayed overlap weight:

$$w_{(j,i)} = \min\left(nH_j^{(\text{pred})}, nH_i^{(\text{true})}\right) \exp\left(-\frac{1}{2}\left(\frac{|\widehat{\delta}_j^{(H)} - \delta_i^{(H)}|}{\sigma}\right)^2\right), \quad \sigma = 0.06 \text{ ppm}.$$

The pair $(j, i)$ with the largest $w_{(j,i)}$ is retained whenever $|\widehat{\delta}_j^{(H)} - \delta_i^{(H)}| \leq \tau_H$. Define $W_{\text{match}} = \sum_{(j,i) \in \widehat{M}} w_{(j,i)}$ as the matched weight, $W_{\text{pred}} = \sum_{j=1}^{N_{\text{pred}}} nH_j^{(\text{pred})}$ and $W_{\text{true}} = \sum_{i=1}^{N_{\text{true}}} nH_i^{(\text{true})}$ as the

total proton counts. We report the *weighted Jaccard* similarity:

$$\text{Jac} = \frac{W_{\text{match}}}{W_{\text{pred}} + W_{\text{true}} - W_{\text{match}}} \in [0,1],$$

together with unweighted peak-level precision, recall, F1, and mean absolute error (MAE):

$$P = \frac{|\widehat{M}|}{N_{\text{pred}}}, \quad R = \frac{|\widehat{M}|}{N_{\text{true}}}, \quad \text{F1} = \frac{2PR}{P+R}, \quad \text{MAE} = \frac{1}{|\widehat{M}|} \sum_{(j,i)\in\widehat{M}} \left| \widehat{\delta}_j^{(H)} - \delta_i^{(H)} \right|.$$

*Implementation note.* For $^{13}$C, we employ tolerance-based greedy nearest-neighbor matching (unweighted). For $^1$H, we adopt a tolerance-constrained greedy matching that maximizes the Gaussian-overlap weight. Multiplicity annotations (e.g., s/d/t) are parsed but excluded from scoring.

**(1.3) IR and MS Spectrum Generation.** IR and MS spectra are converted to $K$-dimensional real-valued vectors $\mathbf{p}, \mathbf{q} \in \mathbb{R}^K$ and evaluated using cosine similarity:

$$\text{CosSim}(\mathbf{p}, \mathbf{q}) = \frac{\mathbf{p}^\top \mathbf{q}}{\|\mathbf{p}\|_2 \cdot \|\mathbf{q}\|_2}.$$

In summary, our spectrum assessment metrics rigorously evaluate peak-level fidelity for NMR and distributional similarity for IR and MS. These protocols provide modality-specific criteria that complement exact-match and structural metrics. Additional derivations, symbol definitions, and implementation details can be found in Appendix A.3.5.

## 3.4 3D STRUCTURE GENERATION

Molecules are inherently three-dimensional, yet commonly used representations such as SMILES encode only 2D connectivity with very limited stereochemical information. As a result, string-based generative models often fail to capture the full spatial arrangement of atoms or to account for conformational diversity. This limitation leads to well-known issues such as invalid structures, duplicated molecules, or incorrect stereochemistry unless additional constraints are imposed. Consequently, generating accurate 3D structures remains a major challenge for molecular modeling.

To address this challenge, MolSpectLLM is explicitly designed to generate 3D molecular structures in addition to interpreting spectra. During pretraining, we construct a unified textual description that integrates atomic coordinates, atom types, and bond connectivity, enabling the model to jointly learn connectivity

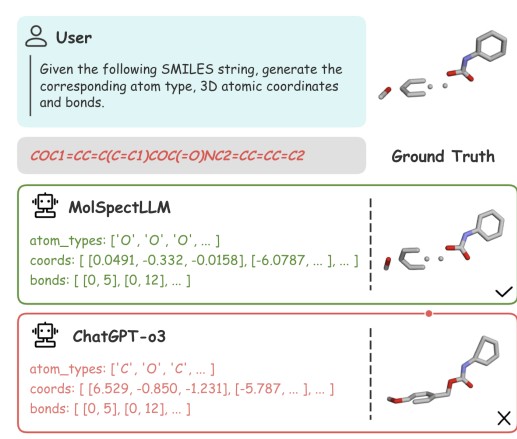

Figure 3: Exmaple of SMILES-to-3D. Mol-SpectLLM is able to generate accurate 3D structure based on the given SMILES string.

and geometry. In the supervised fine-tuning stage, the model takes symbolic inputs (e.g., IUPAC names or SMILES strings) and outputs a complete 3D structure including atomic coordinates and bonding information, as illustrated in Fig. 3. Furthermore, MolSpectLLM can directly leverage multiple experimental spectra: it first predicts a SMILES representation from spectral signals and then generates the corresponding 3D conformation, thereby establishing an end-to-end pathway from raw spectra to spatial molecular structures.

By incorporating 3D structure generation, MolSpectLLM can capture molecular shape and stereochemistry that are essential for understanding reactivity and interactions (Platzer et al., 2025). Unlike 2D or string-only models, this design allows direct optimization of spatial properties and yields physically plausible conformations that respect chemical constraints (Baillif et al., 2023). This capability is particularly important in computational chemistry and drug discovery, where accurate conformations underpin tasks such as binding affinity prediction and structure-based design (Huang et al.,

2022; Zhang et al., 2023). In summary, the integration of 3D generation enables MolSpectLLM to move beyond symbolic molecular representations and leverage geometry as a first-class signal for downstream applications.

### 3.5 THREE-PHASE LEARNING

As shown in Fig. 1, training of MolSpectLLM is organized into the following three stages:

**Pre-training.** We begin by pretraining on 10M publicly available chemistry papers to endow the model with broad chemical knowledge. Next, we construct unified molecular descriptions by integrating multiple sources of structural and spectroscopic data. Specifically, we collect molecular properties and structural information from PubChem (Kim et al., 2023), simulated Raman and UV spectra from QM9S (Zou et al., 2023), and experimental NMR, IR, and Mass spectra from the Multimodal Spectroscopic Dataset (Alberts et al., 2024). These heterogeneous modalities are converted into standardized textual descriptions that combine 3D coordinates, atom and bond information, and spectrum-specific annotations, enabling the model to jointly learn structural, molecular property, and spectral information.

**Multi-task Mixed Supervised Fine-tuning (SFT).** Building on the unified textual description of molecules, we design a broad set of question–answer style tasks that address several key molecular applications. To begin with, in the task of *3D structure generation*, the model learns to produce atomic coordinates, atom types, and bond connectivity for a given molecule, thereby recovering spatially accurate conformations. In addition, for *spectral analysis*, the model is trained to interpret spectral signals such as NMR, IR, or Mass spectra and to provide chemically meaningful insights, for instance by identifying functional groups or structural motifs associated with characteristic peaks. Moreover, in the task of *molecular name conversion*, the model translates between different chemical notations, converting IUPAC names into SMILES representations and vice versa, which ensures consistent canonicalization across naming systems. Finally, in *spectrum generation*, the model takes a molecular representation as input and predicts the corresponding spectra in a standardized textual format, making it possible to directly evaluate the fidelity of spectral predictions. Together, these tasks align the pretrained knowledge with practical objectives and significantly enhance the model's ability to reason over multimodal chemical inputs.

**Instruction-following SFT.** Consistent with prior observations, full-parameter fine-tuning can erode instruction adherence even when downstream data are benign (Qi et al., 2024; Lyu et al., 2024). To mitigate this degradation while retaining task competence, we adopt parameter-efficient adaptation via LoRA, which updates small low-rank adapters while keeping base weights frozen (Hu et al., 2022). Such lightweight tuning has been shown to better preserve alignment and reduce catastrophic overwriting compared with full fine-tuning (Biderman et al., 2024). Concretely, we apply LoRA on a small set of template-aligned examples solely for evaluation formatting and task phrasing. To safeguard the integrity of assessment, the data used here are strictly disjoint from both the post-training SFT corpus and all evaluation sets, ensuring no data leakage.

## 4 EXPERIMENTS

### 4.1 EXPERIMENTAL SETUP

**Tasks.** We evaluate our MOLSPECTLLM model on a diverse set of downstream tasks, including molecular question answering (QA), name conversion, 3D coordinate generation, and molecular spectra generation. In the molecular QA task, the model is required to reason over molecular representations such as SMILES, IUPAC names, or molecular formulas. The name conversion task assesses the ability to translate between SMILES and IUPAC names. For 3D coordinate generation, the model takes SMILES or IUPAC names as input and produces the corresponding 3D molecular structures. Finally, molecular spectra tasks are divided into two categories: *Spectra-to-SMILES*, where the model predicts SMILES representations from multiple given spectra (including IR, NMR, and MS), and *SMILES-to-Spectra*, where the model generates molecular spectra from a given SMILES string.

**Dataset and preprocessing.** During pretraining, we pre-train our model on 10M public chemistry papers, molecular description data built on PubChem (Kim et al., 2023), molecular spectra

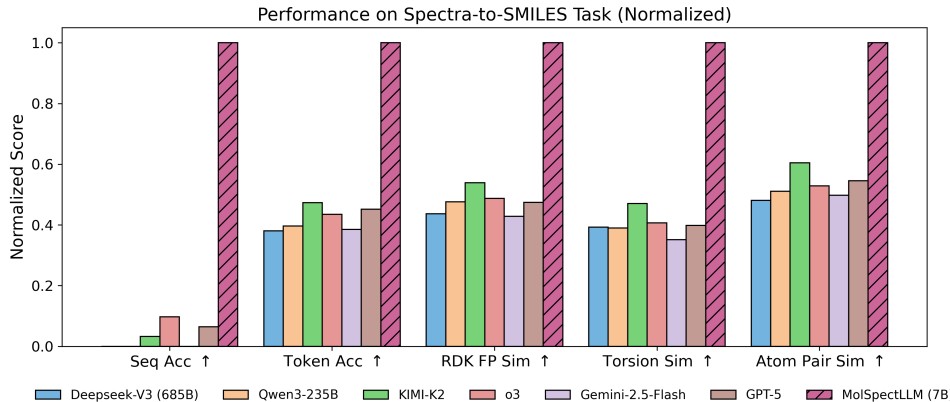

Figure 4: Results on the *Spectra-to-SMILES* task with evaluation metrics including token accuracy, sequence accuracy, FP similarity, and structural similarity.

description data built on NMRBank (Wang et al., 2025), Multimodal Spectroscopic Dataset (Alberts et al., 2024), QM9S (Zou et al., 2023). We filter out duplicated or corrupted molecules, as well as molecular spectra with insufficient signal-to-noise ratios from the dataset. In the end, we used approximately 5M molecules and 0.2M spectra data to train our model. And more details can be found in Sec. A.1.

**Baselines** contain several state-of-the-art multimodal LLMs in general domain, including DeepSeek-V3 (DeepSeek-AI, 2024), Qwen3-235B (Qwen Team, 2025), Kimi-K2 (Kimi Team et al., 2025), OpenAI o3 (OpenAI, 2025b), Gemini-2.5-Flash (Gemini Team, 2025), and GPT-5 (OpenAI, 2025a).

**Implementation Details** are elaborated in Sec A.3.1 in Appendix.

Table 1: Results on *SMILES-to-Spectra* across four spectrum types with similarity metrics. On each task, the best model is **bolded**.

| Model | $^{13}$C-NMR | | $^1$H-NMR | | | IR | MS |
|---|---|---|---|---|---|---|---|
| | F1 ($\uparrow$) | MAE ($\downarrow$) | Jaccard ($\uparrow$) | F1 ($\uparrow$) | MAE $\downarrow$ | CosSim ($\uparrow$) | CosSim ($\uparrow$) |
| Deepseek-V3 (685B) | 0.204 | 0.226 | 0.209 | 0.526 | 0.053 | 0.140 | 0.021 |
| Qwen3-235B | 0.200 | 0.227 | 0.162 | 0.426 | 0.052 | 0.164 | 0.046 |
| KIMI-K2 | 0.254 | 0.219 | 0.216 | 0.513 | 0.052 | 0.150 | 0.074 |
| o3-mini | 0.186 | 0.238 | 0.164 | 0.419 | 0.051 | 0.095 | 0.185 |
| Gemini-2.5-Flash-Lite | 0.172 | 0.234 | 0.153 | 0.471 | 0.055 | 0.146 | 0.026 |
| GPT-5-mini | 0.212 | 0.226 | 0.223 | 0.495 | 0.048 | 0.149 | 0.223 |
| MolSpectLLM (7B) | **0.479** | **0.149** | **0.449** | **0.658** | **0.033** | **0.554** | **0.423** |

## 4.2 EVALUATION METRICS

Our evaluation protocol covers both spectrum-related and structure-related tasks. Spectrum-specific metrics for NMR, IR, and MS generation are introduced in detail in Sec. 3.3, with further derivations and implementation details provided in Appendix A.3.5. Here, we briefly summarize the remaining metrics used throughout our experiments.

**Token- and Sequence-level Accuracy.** For sequence generation tasks (e.g., SMILES–IUPAC conversion), *Token Accuracy* measures the fraction of correctly predicted tokens, while *Sequence Accuracy* reports the proportion of exactly matched sequences (up to canonicalization).

**3D Structure Validity and Geometry.** For coordinate generation tasks, we assess validity and plausibility using: (i) *SDF Validity*, the percentage of parsable molecules; (ii) *Atom Clash*, the average number of severe steric overlaps; (iii) *Bond Violation*, the average number of abnormal bond lengths.

**Fingerprint Similarity.** To quantify structural similarity, we compute Tanimoto coefficients over RDKit fingerprints, including path-based, topological torsion, and atom-pair fingerprints.

Morevoer, spectrum fidelity metrics are detailed in Sec. 3.3, while textual accuracy, structural validity, and molecular similarity are evaluated with the above metrics. Full definitions and additional details are provided in Appendix A.3.5.

### 4.3 RESULTS & ANALYSIS

#### 4.3.1 *Spectra-to-SMILES*

For this task, MolSpectLLM achieves consistent and substantial improvements across all evaluation metrics, as summarized in Figure 4 and Table 3. In terms of sequence-level performance, precision improves from 1.50% to 15.50%, and token-level accuracy rises from 19.73% with KIMI-K2 to 41.65%. Beyond discrete accuracy, structural fidelity is also enhanced, with RDKit Fingerprint similarity increasing from 0.247 to 0.458, Topological Torsion from 0.169 to 0.359, and Atom-Pair from 0.278 to 0.460. As shown in Fig. 5, MolSpectLLM can jointly analyze heterogeneous spectral modalities (e.g., NMR, IR, and MS), extract complementary structural clues from each, and synthesize them into a coherent SMILES prediction, demonstrating its capability to reason over multi-modal spectroscopic evidence rather than relying on any single spectrum. Collectively, these results show that conditioning on molecular spectra substantially mitigates structural ambiguity and enables MolSpectLLM to establish a reliable mapping from spectral signatures to chemically valid and topologically consistent molecular structures. Additional experimental results and analyses are provided in Sec. A.4.1.

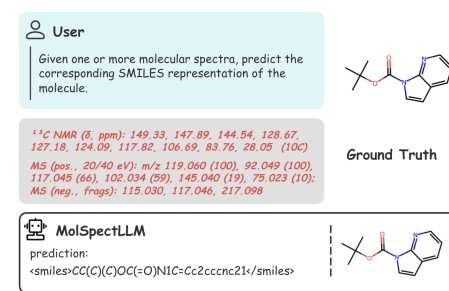

Figure 5: Example of *Spectra-to-SMILES*. MolSpectLLM infers the corresponding molecular SMILES from **multiple** given spectra.

#### 4.3.2 *SMILES-to-Spectra*

MolSpectLLM achieves the strongest performance on spectral generation, as shown in Tab 1 and Fig 11 . For $^{13}$C NMR, the F1 score increases from 0.254 with the best baseline to 0.479 with MolSpectLLM, representing an improvement of nearly 90%, while the MAE decreases from 0.238 to 0.149, marking the lowest error among all models. For $^{1}$H NMR, MolSpectLLM attains the highest Jaccard similarity of 0.449 and the lowest MAE of 0.033. For IR and MS, cosine similarity improves substantially, rising from 0.164 to 0.554 for IR and from 0.223 to 0.423 for MS. As shown in Fig. 6, MolSpectLLM not only excels at interpreting spectra to recover molecular structures, but also predicts spectra from structure with higher fidelity across multiple modalities.

#### 4.3.3 *Name Conversion*

On the SMILES-to-IUPAC task, MolSpectLLM achieves 78.59% token accuracy and 54.05% sequence accuracy as shown in Tab. 2. This represents a substantial improvement over previous models, where the best baseline, KIMI-K2, reached only 11.60% token accuracy and the o3 model achieved 2.00% sequence accuracy. Thus, MolSpectLLM improves token-level performance by nearly seven times and sequence-level accuracy by more than twenty-five times, as illustrated in

Table 2: Results on MolQA, SMILES-to-IUPAC, and IUPAC-to-SMILES with token- and sequence-level accuracy (**best**).

| Model | MolQA Acc (↑) | SMILES-to-IUPAC | | IUPAC-to-SMILES | |
|---|---|---|---|---|---|
| | | Token Acc (↑) | Seq Acc (↑) | Token Acc (↑) | Seq Acc (↑) |
| Deepseek-V3 (685B) | 54.60 | 6.94 | 0.00 | 48.83 | 12.00 |
| Qwen3-235B | 52.00 | 6.78 | 0.00 | 27.60 | 0.00 |
| KIMI-K2 | 53.20 | 11.60 | 1.00 | 53.51 | 22.00 |
| o3 | 81.20 | 10.86 | 2.00 | 47.09 | 18.00 |
| Gemini-2.5-Flash | 72.20 | 11.09 | 1.00 | 48.37 | 13.98 |
| GPT-5 | **88.20** | 10.45 | 1.00 | 51.04 | 28.00 |
| MolSpectLLM (7B) | 67.00 | **78.59** | **54.05** | **72.54** | **66.72** |

Tab. 2. For the inverse IUPAC-to-SMILES task, MolSpectLLM attains 72.54% token accuracy and 66.72% sequence accuracy. In comparison, the strongest baseline, KIMI-K2, obtained 53.51% token accuracy, while GPT-5 reached only 28.00% sequence accuracy. These results indicate that MolSpectLLM can reliably handle chemically consistent canonicalization and learn the non-trivial bijective mappings between notations, whereas large general-purpose models produce almost no exact matches.

### 4.3.4 *Molecule QA*

As shown in Tab. 2, in the molecular elucidation task, MolSpectLLM demonstrates strong capabilities, though it does not yet surpass the performance of larger closed-source models with broader knowledge bases. Nevertheless, compared to most open-source alternatives, MolSpectLLM remains highly competitive. In particular, relative to its backbone Qwen series, MolSpectLLM achieves an accuracy of 67%, substantially outperforming Qwen3-235B (52%). These results highlight the benefits of integrating spectroscopy and 3D structural reasoning, enabling MolSpectLLM to deliver significant gains within the open-source model landscape.

### 4.3.5 *3D structure generation*

On the SMILES-to-3D task, MolSpectLLM achieves the highest structural validity at 89.68% and the best topological agreement with a fingerprint similarity of 0.582. Compared to GPT-5, which attains 69.50% validity, and o3, which reaches 0.356 similarity, these results reflect clear improvements in both reliability and fidelity. Although GPT-5 and KIMI-K2 report fewer clashes and bond violations, their valid outputs are far less frequent and structurally consistent, underscoring that low error counts on limited subsets can be misleading. Overall, MolSpectLLM produces more valid and faithful structures, while still leaving scope to re-

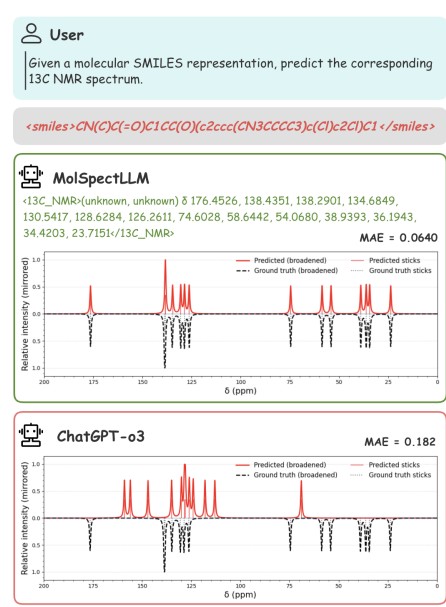

Figure 6: Example of *SMILES-to-Spectra*. MolSpectLLM generates chemically consistent spectra from a given SMILES. Here, it accurately predicts the $^{13}$C NMR spectrum with a mean error of only 0.064.

duce steric and bonding artifacts. And on the IUPAC-to-3D task, MolSpectLLM again delivers the best validity at 82.78% and the lowest bond-length errors, averaging 1.357 violations compared to 2.059 for GPT-5. Its fingerprint similarity of 0.705 is slightly below the 0.813 achieved by GPT-5, indicating that MolSpectLLM prioritizes geometric accuracy at high validity rates, whereas GPT-5 attains somewhat higher topological overlap.

## 5 CONCLUSION

In this work, we presented MolSpectLLM, a molecular foundation model that unifies experimental spectroscopy with three-dimensional structural generation. By explicitly modeling IR, NMR, MS, and other spectral modalities, MolSpectLLM not only achieves strong performance in individual spectrum interpretation and generation, but also demonstrates the ability to jointly analyze multiple spectra to extract complementary information. Beyond spectroscopy, it performs competitively on benchmarks such as molecule QA and SMILES–IUPAC conversions, and further enables accurate 3D molecular structure generation directly from textual or spectral inputs. These results highlight the value of integrating experimental and structural modalities for advancing molecular understanding and design. Looking ahead, we plan to scale both the model size and training data to further strengthen spectral reasoning, improve instruction alignment, and enhance the balance between molecular expertise and general usability. We believe these directions will pave the way toward more versatile and practically useful molecular foundation models.

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

# A   APPENDIX

## A.1   DATASETS

### A.1.1   PUBCHEM.

PubChem is a large, publicly accessible chemical information database that integrates data from hundreds of sources. As of recent updates, it contains over 119 million unique compounds and aggregates information from more than 1000 data sources (Kim et al., 2023). We leverage PubChem to obtain fundamental molecular identity records, including multiple representations and textual descriptors for each compound. In practice, for each molecule we retrieve its SMILES strings, IUPAC names, molecular formulae, and known synonyms from PubChem, along with any brief descriptive annotations available. These rich, cross-referenced identifiers provide a foundation for tasks like molecular QA (querying chemical facts) and name-to-structure conversion, ensuring that the model can recognize and interconvert between different naming conventions and representations of the same compound. By using the extensive coverage of PubChem, which spans a broad chemical space and connects to many auxiliary data points, we ensure comprehensive molecular identity information is included for pretraining.

### A.1.2   QM9S.

To incorporate high-quality quantum chemical references, we use the **QM9S** dataset (Zou et al., 2023). QM9S is an augmented version of the popular QM9 dataset of small organic molecules (up to 9 non-hydrogen atoms). It consists of about 130,000 molecules (composed of C, H, N, O, F) derived from QM9, for which the geometries and properties have been recomputed at a higher level of theory. Specifically, Zou et al. optimized each molecule's 3D structure with DFT (B3LYP/def-TZVP) and then calculated a wide range of physico-chemical properties, including thermodynamic energies, partial charges, dipole moments, higher-order multipole moments, polarizabilities, and other tensorial properties. Importantly, they also simulated several types of spectra from first principles: frequency analysis and time-dependent DFT computations were used to generate **infrared (IR)** and **Raman** spectra, as well as **UV-Vis** absorption spectra for each molecule. This corpus thus offers chemically consistent 3D structures paired with theoretically calculated spectral data. In our training, we use QM9S both to teach the model about accurate molecular geometries and to enable *spectrum simulation tasks* under ideal conditions.

### A.1.3   NMRBANK.

For experimental spectroscopic data, we draw from **NMRBank**, a recently curated collection of nuclear magnetic resonance records built from the chemical literature (Wang et al., 2025). Wang et al. constructed NMRBank by using a language-model-based text mining tool named NMRExtractor to process over 5.7 million scientific publications. The result is a database of about **225,809 entries** of compounds with their reported $^1$H and $^{13}$C NMR chemical shifts, along with metadata such as the experimental conditions (solvent, spectrometer frequency, etc.), confidence indicators, and reference citations. This offers an unprecedented scale of real-world NMR information, far surpassing older public NMR datasets in chemical diversity and size. We include NMRBank in our pretraining corpus to expose the model to genuine experimental spectra characteristics – for instance, the typical chemical shift ranges for various functional groups and the variability of NMR data across different molecules. By retaining the linkage between each NMR record and its compound, the model can learn to associate structural features with NMR signatures (and vice versa) in a realistic context.

### A.1.4   MULTIMODAL SPECTROSCOPIC DATASET.

In addition to NMRBank, we incorporate a broad **multimodal spectroscopic dataset** introduced by Alberts et al.. This dataset – one of the first of its kind – provides **simulated spectra across six different spectroscopic techniques** for approximately **790,000 organic molecules** extracted from reaction outcomes in patent databases. For each molecule, the dataset includes predicted spectra or spectral features: $^1$H NMR, $^{13}$C NMR, HSQC NMR (a 2D technique), infrared (IR) absorption, and tandem mass spectrometry (MS/MS) in both positive and negative ionization modes. All spectra are computationally simulated; for example, NMR peaks and shifts are predicted, IR intensities

are generated over standard frequency ranges, and MS/MS data list fragment peaks with putative fragment formulas. Despite being synthetic, the dataset is designed to reflect realistic experimental outputs. By training on this multimodal dataset, our model learns to handle **multiple spectroscopic modalities in combination**, mirroring how chemists use complementary techniques for structure elucidation.

### A.1.5  COMPUTED VS. EXPERIMENTAL SPECTRA.

It is important to note the differences between **computed** spectral data like QM9S and **experimental or realistic** spectral data such as NMRBank and the Multimodal Spectroscopic Dataset. QM9S provides high-quality, physics-based data generated under uniform theoretical conditions – highly consistent and reproducible, but lacking the variability of laboratory conditions such as solvent effects or instrument noise. In contrast, NMRBank entries and the patent-derived multimodal dataset embody the complexity of real-world chemistry. The multimodal dataset's spectra, although simulated, cover a broad range of molecular size and functional complexity, while NMRBank provides true experimental chemical shifts, inherently including condition-dependent variations. By combining these sources, we ensure that the model learns both idealized theoretical patterns and pragmatic, experimentally relevant spectra, improving robustness across both spectrum-to-structure and structure-to-spectrum tasks.

### A.2  DATA PROCESSING

### A.2.1  PUBCHEM.

For molecular identity and descriptor information, we curated a large-scale dataset from the PubChem compound archive, which provides both 2D and 3D SDF files for millions of compounds. We processed these files using the RDKit cheminformatics toolkit to extract a comprehensive set of molecular features. Each molecule is indexed by its PubChem Compound ID (CID), and all parsed records are stored in both a dictionary (CID↦features) and an indexed list for efficient retrieval.

**2D information.**  From the 2D SDF files, we extracted the following fields explicitly provided by PubChem: canonical SMILES strings, molecular formulae, molecular weight, exact mass, heavy atom count, rotatable bond count, H-bond donors/acceptors, and associated identifiers. In addition, we recorded approximate 2D coordinates of atoms (when present in the file) for visualization or graph layout purposes. These descriptors cover basic chemical identity and structural properties.

**3D information.**  From the 3D SDF files, we used RDKit to obtain a full set of atomic- and molecular-level descriptors:

- *Atomic-level features:* atom indices, atom types (element symbols), formal charges, aromaticity flags, chirality tags (whether an atom has a specified stereochemical label), ring membership (atoms in rings), 3D Cartesian coordinates (from conformers), and explicit bond connectivity (pairs of atom indices) with bond order (single/double/triple).

- *Ring structures:* list of the smallest set of smallest rings (SSSR) per molecule, allowing enumeration of aromatic and non-aromatic rings.

- *Electrostatic descriptors:* per-atom Gasteiger charges, which approximate atomic partial charges from electronegativity equalization.

**Molecular fingerprints.**  Several widely used RDKit fingerprints were computed for each molecule:

- **MACCS keys:** a 166-bit structural key fingerprint capturing presence/absence of common substructures.

- **RDKit fingerprint:** a path-based hashed fingerprint enumerating atom-bond paths.

- **E-State fingerprint:** electrotopological state fragment counts.

**Physicochemical descriptors.** We also computed common molecular descriptors from RDKit's `Descriptors` module:

- Number of valence electrons (`NumValenceElectrons`);
- Topological polar surface area (TPSA);
- Octanol-water partition coefficient (MolLogP).

**Feature annotations.** Using RDKit's feature factory (`ChemicalFeatures`), we enumerated pharmacophore-like features such as hydrogen bond donors, acceptors, aromatic centers, and hydrophobic groups. These annotations provide higher-level semantic features for each molecule.

**Data organization.** The processed dataset thus contains, for each PubChem compound: (i) identifiers and basic properties from the raw SDF (CID, SMILES, formula, exact mass, etc.); (ii) 2D coordinates and counts of functional features (donors, acceptors, rotatable bonds); (iii) full 3D atomic and bonding information; (iv) aromaticity, chirality, and ring structures; (v) multiple types of fingerprints; (vi) physicochemical descriptors and atomic partial charges; and (vii) higher-level chemical features.

Finally, we selected some features suitable for use as language model input. These features form a unified molecular textual description combining identity, structural, electronic, and pharmacophoric information for millions of compounds, enabling downstream molecular QA, name conversion, and 3D generation tasks.

### A.2.2 SPECTRUM DATA

**$^1$H NMR Spectroscopy.** Proton nuclear magnetic resonance ($^1$H NMR) spectroscopy exploits the magnetic properties of hydrogen nuclei to probe molecular structure. Protons in different chemical environments resonate at characteristic frequencies (chemical shifts, $\delta$ in ppm), which reflect electron shielding effects (Keeler, 2011). Multiplicity arises from spin–spin coupling with neighboring hydrogens, quantified by coupling constants ($J$ in Hz), while integration reveals the number of protons contributing to each signal (Claridge, 2016). These features provide detailed insights into functional groups and connectivity. We translate raw spectral vectors into structured textual descriptions capturing chemical shifts, multiplicities, couplings, and integrations, thereby embedding human-interpretable NMR cues in a form amenable to LLM-based reasoning.

---

**Algorithm 1** Textual conversion for $^1$H NMR

---

**Require:** Raw vector of $^1$H NMR peaks: $\{\delta_i, n_i, m_i, J_i\}$
**Ensure:** Formatted textual representation
1: Initialize $rep \leftarrow$ "`<1H_NMR>(frequency, solvent)`"
2: **for** each peak $i$ **do**
3:     Format shift $\delta_i$ (ppm)
4:     Extract multiplicity $m_i$ and integration $n_i$
5:     **if** coupling constants $J_i$ available **then**
6:         Append "J = ... Hz"
7:     **end if**
8:     Append to $rep$: "$\delta_i$ ($m_i, n_i$H)"
9: **end for**
10: Close tag: $rep \leftarrow rep+$ "`</1H_NMR>`"
11: **return** $rep$

---

**$^{13}$C NMR Spectroscopy.** Carbon-13 NMR ($^{13}$C NMR) provides a complementary view of molecular skeletons. $^{13}$C chemical shifts span a wide range (0–220 ppm), diagnostic of hybridization and functional groups: sp$^3$ carbons at 0–50 ppm, sp$^2$ aromatic carbons around 110–160 ppm, and carbonyl carbons beyond 160 ppm (Claridge, 2016). Unlike $^1$H NMR, broadband-decoupled $^{13}$C spectra usually display single peaks per carbon environment without multiplicities, and intensities are not strictly quantitative (Keeler, 2011). Translating spectra into textual form involves listing chemical shift values and identifying characteristic regions (carbonyl, aromatic, aliphatic).

---

**Algorithm 2** Textual conversion for $^{13}$C NMR

---

**Require:** Raw vector of $^{13}$C NMR shifts: $\{\delta_i\}$
**Ensure:** Formatted textual representation
1: Sort $\delta_i$ values descending
2: Initialize $rep \leftarrow$ "`<13C_NMR>(frequency, solvent) δ`"
3: **for** each shift $\delta_i$ **do**
4:     Append to $rep$: "$\delta_i$"
5: **end for**
6: Close tag: $rep \leftarrow rep+$ "`</13C_NMR>`"
7: **return** $rep$

---

**Infrared Spectroscopy.** Infrared spectroscopy probes vibrational transitions of chemical bonds. Characteristic absorption bands correspond to functional groups: broad O–H stretches at 3200–3600 cm$^{-1}$, C=O carbonyl stretches at 1650–1800 cm$^{-1}$, C–H stretches near 2850–3000 cm$^{-1}$, and sharp nitrile bands at 2250 cm$^{-1}$ (Colthup, 2012; Smith, 2018). By extracting peak positions and intensities, we generate textual summaries indicating functional group assignments.

---

**Algorithm 3** Textual conversion for IR Spectrum

---

**Require:** Raw IR spectrum: frequency–intensity pairs $\{(\nu_i, I_i)\}$
**Ensure:** Formatted textual representation
1: Identify peaks above threshold
2: Initialize $rep \leftarrow$ "`<IR>(500~4000)`"
3: **for** each peak $(\nu_i, I_i)$ **do**
4:     Append to $rep$: "$\nu_i$ ($I_i$)"
5: **end for**
6: Close tag: $rep \leftarrow rep+$ "`</IR>`"
7: **return** $rep$

---

**Mass Spectrometry.** Mass spectrometry (MS) measures mass-to-charge ($m/z$) ratios of ions, providing molecular weight and fragmentation patterns. The molecular ion ($M^+$) reveals molecular mass, while fragment ions (e.g. tropylium at m/z 91, phenyl cation at m/z 77) indicate structural motifs (Gross, 2017). Textual conversion enumerates major peaks and their intensities, normalized to the base peak (100%).

---

**Algorithm 4** Textual conversion for Mass Spectrum

---

**Require:** List of peaks $\{(m/z_i, I_i)\}$, normalized to base peak=100%
**Ensure:** Formatted textual representation
1: Sort peaks by $m/z$
2: Initialize $rep \leftarrow$ "`<ms_positive>`"
3: **for** each peak $(m/z_i, I_i)$ **do**
4:     Append to $rep$: "$m/z_i : I_i$"
5: **end for**
6: Close tag: $rep \leftarrow rep+$ "`</ms_positive>`"
7: **return** $rep$

---

### A.2.3  INSTRUCTION DATA

Based on the processed features obtained from our datasets, we constructed a large collection of instruction-tuning data. As illustrated in Fig. 1, these data cover a diverse set of tasks:

- **Molecule QA:** question–answer pairs targeting both local and global molecular features.
- **Structure Generation:** the model is required to generate 3D structural coordinates together with atom types and bond types.

- **IUPAC to SMILES:** the model is asked to convert a given IUPAC name into its corresponding SMILES string.
- **SMILES to IUPAC:** the model is asked to generate an IUPAC name from a given SMILES representation.
- **Spectrum to SMILES:** given one or multiple standard textual descriptions of spectra, the model is required to output the corresponding molecular SMILES.
- **SMILES to Spectrum:** given a molecular SMILES, the model is required to predict a specified spectrum in textual form.

Each task is instantiated in multiple QA formats, including free-form question answering, multiple-choice questions, and true/false judgments. Importantly, all QA templates used here are distinct from those employed in the Instruction-Following SFT stage, ensuring no overlap between training and evaluation templates and thus mitigating overfitting and data leakage.

### A.3    METHOD DETAILS

#### A.3.1    IMPLEMENTATION DETAILS

All experiments are conducted on a single node with 8 NVIDIA A800 GPUs. During training, the sequence length is truncated to a maximum of 4096 tokens. The model is trained with a per-device batch size of 4 and a gradient accumulation step of 8, yielding an effective batch size of 32. We employ a learning rate of $1.0 \times 10^{-5}$ with a cosine learning rate scheduler and a warm-up ratio of 0.1.

#### A.3.2    BASE MODEL

We primarily choose **Qwen2.5-7B** (Qwen et al., 2025) as the base model architecture. Qwen2.5-7B is a 7-billion-parameter transformer-based language model, featuring a decoder-only architecture with multihead self-attention and rotary position embeddings. The model was pretrained on a large-scale mixed-domain corpus spanning web documents, code, and scientific texts, which endows it with strong general-purpose language understanding and generation capabilities. Compared to smaller variants, the 7B model strikes a balance between scalability and efficiency, offering sufficient parameter capacity to capture complex multimodal patterns while remaining feasible for fine-tuning on our spectroscopy–structure tasks.

#### A.3.3    PRETRAINING

As shown in Fig. 7, we pretrain MolSpectLLM for one epoch on our unified molecular textual description dataset.

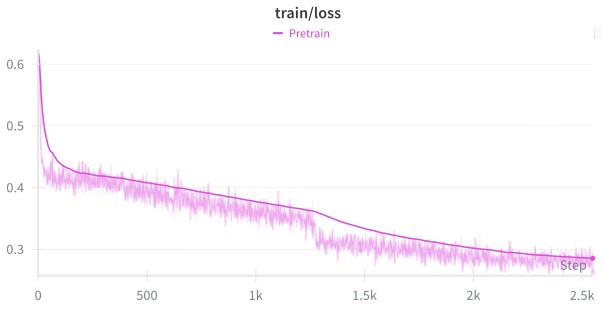

Figure 7: Training loss curve of pretraining.

#### A.3.4    MULTI-TASK MIXED SFT

As shown in Fig. 8, we fine-tune the pretrained MolSpectLLM on all kinds of instrcution data based on the unified molecular textual descriptions for three epochs.

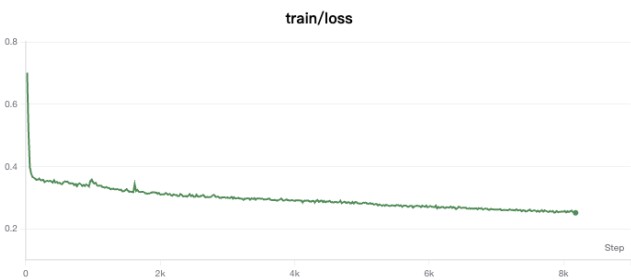

Figure 8: Training loss curve of Multi-task Mixed SFT.

### A.3.5 EVALUATION

**Token-level and Sequence-level Accuracy.** For sequence generation tasks (e.g., SMILES to IU-PAC), let the test set be $\mathcal{D} = \{(T_i, \hat{T}_i)\}_{i=1}^N$, where $T_i = (t_{i,1}, \ldots, t_{i,m_i})$ is the ground-truth token sequence and $\hat{T}_i = (\hat{t}_{i,1}, \ldots, \hat{t}_{i,n_i})$ is the model output. We use a canonicalization map $\mathcal{C}(\cdot)$ (e.g., canonical SMILES) applied to whole sequences before exact comparison. The indicator $\mathbf{1}\{\cdot\}$ returns 1 if the condition holds and 0 otherwise.

**Token Accuracy (per-sample).**

$$\text{TokenAcc}(T_i, \hat{T}_i) = \frac{1}{|T_i|} \sum_{j=1}^{|T_i|} \mathbf{1}\{\hat{t}_{i,j} = t_{i,j}\}, \quad |T_i| = m_i. \tag{1}$$

$t_{i,j}$ is the $j$-th token of the ground truth for sample $i$; $\hat{t}_{i,j}$ is the $j$-th token of the prediction (if $j > n_i$, we treat $\hat{t}_{i,j}$ as missing and hence mismatched). The reported Token Accuracy is $\frac{1}{N} \sum_{i=1}^N \text{TokenAcc}(T_i, \hat{T}_i)$.

**Sequence Accuracy (dataset-level).**

$$\text{SeqAcc} = \frac{1}{N} \sum_{i=1}^N \mathbf{1}\{\mathcal{C}(\hat{T}_i) \equiv \mathcal{C}(T_i)\}. \tag{2}$$

$\equiv$ denotes exact string equality after canonicalization; $N$ is the number of test samples. A sample contributes 1 iff the entire canonicalized ($\mathcal{C}(\cdot)$) prediction matches the canonicalized ground truth.

---

**Algorithm 5** Compute Token & Sequence Accuracy

---

**Require:** Test set $\mathcal{D} = \{(T_i, \hat{T}_i)\}_{i=1}^N$; canonicalizer $\mathcal{C}(\cdot)$
1: $S \leftarrow 0$                                                      ▷ exact sequence match counter
2: $A \leftarrow 0$                                                   ▷ sum of per-sample token accuracies
3: **for** $i = 1$ to $N$ **do**
4:     $T_i' \leftarrow \mathcal{C}(T_i); \hat{T}_i' \leftarrow \mathcal{C}(\hat{T}_i)$
5:     **if** $\hat{T}_i' = T_i'$ **then**
6:         $S \leftarrow S + 1$
7:     **end if**
8:     $m \leftarrow |T_i|; n \leftarrow |\hat{T}_i|; c \leftarrow 0$
9:     **for** $j = 1$ to $m$ **do**
10:         **if** $j \leq n$ **and** $\hat{t}_{i,j} = t_{i,j}$ **then**
11:             $c \leftarrow c + 1$
12:         **end if**
13:     **end for**
14:     $A \leftarrow A + \frac{c}{m}$
15: **end for**
16: **return** SeqAcc $= \frac{S}{N}$,     TokenAcc $= \frac{A}{N}$

---

**Structure Validity and Geometry Quality.** Let a predicted 3D structure be $M = (G, X)$ where $G = (V, E)$ is the molecular graph ($V$ atoms with element types $z_i$, $E$ bonds with orders $o_{ij}$) and $X \in \mathbb{R}^{|V| \times 3}$ are Cartesian coordinates ($x_i$ for atom $i$).

**SDF Validity.**

$$\text{SDFValid} = \frac{N_{\text{valid}}}{N_{\text{total}}}. \tag{3}$$

$N_{\text{valid}}$ is the count of predictions that can be parsed into chemically valid molecules by a toolkit (format OK, valence reasonable, non-empty graph), $N_{\text{total}}$ is the number of generated files.

**Atom Clash (steric overlap).** Define the set of non-bonded pairs $\mathcal{P}_{\text{nb}}(G) = \{(i, j) : i < j, \ (i, j) \notin E\}$. Let $d_{ij} = \|x_i - x_j\|_2$ and $r_i^{\text{vdW}}$ be the element-wise van der Waals radius. A clash occurs if

$$d_{ij} < \alpha \left(r_i^{\text{vdW}} + r_j^{\text{vdW}}\right), \quad \alpha = 0.65. \tag{4}$$

The reported Atom Clash is the average number of clashing pairs per molecule.

**Bond Violation (length out-of-range).** For a bonded pair $(i, j) \in E$ with order $o_{ij}$, let the reference length be $\ell_{ij}^{(o_{ij})}$ from a lookup table conditioned on $(z_i, z_j, o_{ij})$. A violation occurs if

$$d_{ij} \notin \left[(1 - \beta) \, \ell_{ij}^{(o_{ij})}, \ (1 + \beta) \, \ell_{ij}^{(o_{ij})}\right], \quad \beta = 0.20. \tag{5}$$

The reported Bond Violation is the average number of violated bonds per molecule.

---

**Algorithm 6** Geometry Diagnostics: SDF Validity, Atom Clash, Bond Violation

---

**Require:** Predicted set $\{M_k = (G_k, X_k)\}_{k=1}^{N_{\text{total}}}$; parser $\text{Parse}(\cdot)$; radii $r^{\text{vdW}}(z)$; reference lengths $\ell(z_i, z_j, o)$; $\alpha = 0.65$, $\beta = 0.20$
1: $V \leftarrow 0; C \leftarrow 0; B \leftarrow 0$
2: **for** $k = 1$ to $N_{\text{total}}$ **do**
3:     **if** $\text{Parse}(M_k)$ succeeds **then**
4:         $V \leftarrow V + 1$
5:     **end if**
6:     $c \leftarrow 0; b \leftarrow 0; (G, X) \leftarrow M_k$
7:     **for all** $(i, j) \in \mathcal{P}_{\text{nb}}(G)$ **do**
8:         $d \leftarrow \|x_i - x_j\|_2$
9:         **if** $d < \alpha \left[r^{\text{vdW}}(z_i) + r^{\text{vdW}}(z_j)\right]$ **then**
10:           $c \leftarrow c + 1$
11:         **end if**
12:     **end for**
13:     **for all** $(i, j, o) \in E$ **do**
14:         $d \leftarrow \|x_i - x_j\|_2; \quad L \leftarrow \ell(z_i, z_j, o)$
15:         **if** $d < (1 - \beta)L$ **or** $d > (1 + \beta)L$ **then**
16:           $b \leftarrow b + 1$
17:         **end if**
18:     **end for**
19:     $C \leftarrow C + c; \quad B \leftarrow B + b$
20: **end for**
21: **return** $\text{SDFValid} = \frac{V}{N_{\text{total}}}$, $\text{AtomClash} = \frac{C}{N_{\text{total}}}$, $\text{BondViolation} = \frac{B}{N_{\text{total}}}$

---

**Fingerprint Similarity.** We compare predicted and reference structures via *Tanimoto similarity* on binary fingerprints. Let $b \in \{0, 1\}^K$ be a fingerprint bit vector and let $|b|_1$ denote its Hamming weight. For two fingerprints $b^{(\text{pred})}, b^{(\text{true})}$:

$$\text{Tanimoto}\left(b^{(\text{pred})}, b^{(\text{true})}\right) = \frac{\langle b^{(\text{pred})}, b^{(\text{true})} \rangle}{|b^{(\text{pred})}|_1 + |b^{(\text{true})}|_1 - \langle b^{(\text{pred})}, b^{(\text{true})} \rangle}. \tag{6}$$

$\langle \cdot, \cdot \rangle$ counts common set bits (intersection size); the denominator is the union size. Values lie in $[0, 1]$.

We report three RDKit (Landrum et al., 2006) fingerprints:

**Path-based (RDKFingerprint; "FP Sim").** Enumerate all simple paths $p = (v_1, \ldots, v_L)$ up to length $L \le L_{\max}$ (typically 7 bonds). Encode a path feature $\phi_{\text{path}}(p)$ from atom types $(z_{v_k})$ and bond types along $p$; hash it to an index $h(\phi) \in \{1, \ldots, K\}$ and set $b_{h(\phi)} \leftarrow 1$.

**Topological Torsion ("Torsion Sim").** Enumerate all sequences of four consecutively bonded atoms $q = (i, j, k, \ell)$ (paths of length 3). Form a torsion feature $\phi_{\text{tor}}(q) = (\tau(z_i), \tau(z_j), \tau(z_k), \tau(z_\ell), \text{bond}_{ij}, \text{bond}_{jk}, \text{bond}_{k\ell})$, where $\tau(\cdot)$ maps raw element/flags to an atom-type class (e.g., element + aromaticity). Hash $\phi_{\text{tor}}$ to set bits. This captures local 4-atom environments (Nilakantan et al., 1987).

**Atom-Pair ("Atom Pair Sim").** For every unordered atom pair $(i, j)$, compute the topological distance $\delta_{ij}$ (shortest path length in $G$). Define an atom-pair feature $\phi_{\text{ap}}(i, j) = (\tau(z_i), \tau(z_j), \delta_{ij})$ and hash to set bits. This captures medium-range topology (Carhart et al., 1985).

---

**Algorithm 7** Fingerprint & Tanimoto Computation

---

**Require:** Molecules $M^{(\text{pred})}, M^{(\text{true})}$; hash $h(\cdot)$; path limit $L_{\max}$; bit length $K$
1: **function** PATHFP($M$)
2:     $b \leftarrow \mathbf{0}_K$
3:     **for all** simple paths $p$ in $M$ with length $\le L_{\max}$ **do**
4:         $\phi \leftarrow \phi_{\text{path}}(p); \quad k \leftarrow h(\phi); \quad b_k \leftarrow 1$
5:     **end for**
6:     **return** $b$
7: **end function**
8: **function** TORSIONFP($M$)
9:     $b \leftarrow \mathbf{0}_K$
10:     **for all** bonded quadruples $q = (i, j, k, \ell)$ in $M$ **do**
11:         $\phi \leftarrow \phi_{\text{tor}}(q); \quad k \leftarrow h(\phi); \quad b_k \leftarrow 1$
12:     **end for**
13:     **return** $b$
14: **end function**
15: **function** ATOMPAIRFP($M$)
16:     $b \leftarrow \mathbf{0}_K$
17:     **for all** unordered pairs $(i, j)$ of atoms in $M$ **do**
18:         $\delta \leftarrow$ shortest-path length between $i$ and $j$ in $G$
19:         $\phi \leftarrow \phi_{\text{ap}}(i, j); \quad k \leftarrow h(\phi); \quad b_k \leftarrow 1$
20:     **end for**
21:     **return** $b$
22: **end function**
23: **function** TANIMOTO($b^{(1)}, b^{(2)}$)
24:     $c \leftarrow \langle b^{(1)}, b^{(2)} \rangle; \quad a \leftarrow |b^{(1)}|_1; \quad b \leftarrow |b^{(2)}|_1$
25:     **if** $a + b - c = 0$ **then**
26:         **return** $0$
27:     **else**
28:         **return** $c/(a + b - c)$
29:     **end if**
30: **end function**
31: $b_{\text{pred}}^{\text{path}} \leftarrow$ PATHFP($M^{(\text{pred})}$); $b_{\text{true}}^{\text{path}} \leftarrow$ PATHFP($M^{(\text{true})}$)
32: $b_{\text{pred}}^{\text{tor}} \leftarrow$ TORSIONFP($M^{(\text{pred})}$); $b_{\text{true}}^{\text{tor}} \leftarrow$ TORSIONFP($M^{(\text{true})}$)
33: $b_{\text{pred}}^{\text{ap}} \leftarrow$ ATOMPAIRFP($M^{(\text{pred})}$); $b_{\text{true}}^{\text{ap}} \leftarrow$ ATOMPAIRFP($M^{(\text{true})}$)
34: **return** FP Sim = TANIMOTO($b_{\text{pred}}^{\text{path}}, b_{\text{true}}^{\text{path}}$), Torsion Sim = TANIMOTO($b_{\text{pred}}^{\text{tor}}, b_{\text{true}}^{\text{tor}}$), Atom Pair Sim = TANIMOTO($b_{\text{pred}}^{\text{ap}}, b_{\text{true}}^{\text{ap}}$)

---

## A.4 ADDITIONAL EXPERIMENTAL RESULTS

In this section, we present supplementary experimental findings and visualizations across different tasks.

### A.4.1 *Spectra-to-SMILES*

Table 3 shows that MolSpectLLM consistently outperforms state-of-the-art (SoTA) general-purpose LLMs on the *Spectra-to-SMILES* task, with substantial gains in both accuracy and fingerprint-based similarity metrics. Figure 9 further illustrates case studies across different spectral combinations, including [1]H-NMR+MS, IR+MS, [13]C-NMR+MS, and paired MS spectra. These examples demonstrate that MolSpectLLM can robustly infer molecular SMILES from diverse spectroscopic evidence.

Table 3: Results on *Spectra-to-SMILES* task with token accuracy, sequence accuracy, fingerprint (FP) similarity, and structural similarity. On each task, the best model is **bolded**.

| Model | Seq Acc (↑) | Token Acc (↑) | RDK FP Sim (↑) | Torsion Sim (↑) | Atom (Pair) Sim (↑) |
|---|---|---|---|---|---|
| Deepseek-V3 (685B) | 0.00 | 15.84 | 0.200 | 0.141 | 0.221 |
| Qwen3-235B | 0.00 | 16.51 | 0.218 | 0.140 | 0.235 |
| KIMI-K2 | 0.50 | 19.73 | 0.247 | 0.169 | 0.278 |
| o3 | 1.50 | 18.12 | 0.223 | 0.146 | 0.243 |
| Gemini-2.5-Flash | 0.00 | 16.02 | 0.196 | 0.126 | 0.229 |
| GPT-5 | 1.00 | 18.82 | 0.217 | 0.143 | 0.251 |
| MolSpectLLM (7B) | **15.50** | **41.65** | **0.458** | **0.359** | **0.460** |

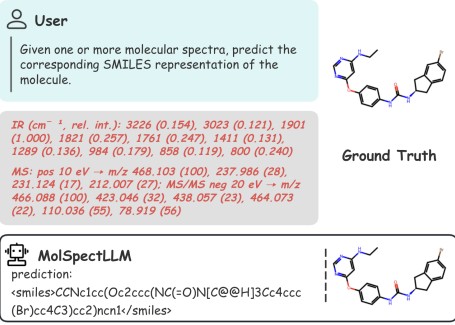

(a) IR & Mass Spectra to SMILES.

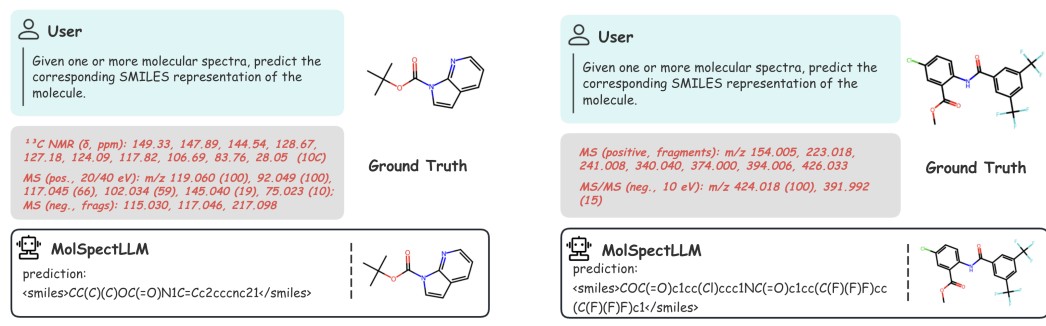

(b) [13]C-NMR & Mass Spectra to SMILES.      (c) Two different Mass Spectra to SMILES.

Figure 9: Case studies of Spectra-to-SMILES on four spectroscopic modalities: ([1]H-NMR, [13]C-NMR, IR, and MS).

### A.4.2 *SMILES-to-Spectra*

We further evaluate MolSpectLLM on the *SMILES-to-Spectra* task, where the model is required to generate spectroscopic representations directly from molecular SMILES. Figure 10 presents representative case studies across three modalities, including mass spectrometry, IR, and [1]H-NMR. In each case, MolSpectLLM produces spectra that closely match the ground truth, capturing both peak positions and relative intensities. These results highlight the model's ability to learn meaningful mappings from structural representations to diverse experimental observables.

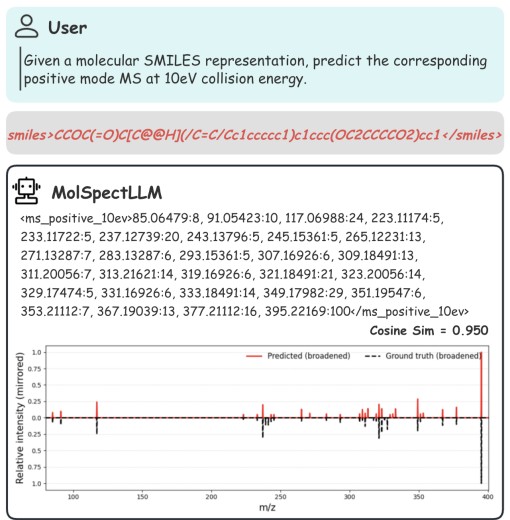

(a) SMILES to Mass spectrum.

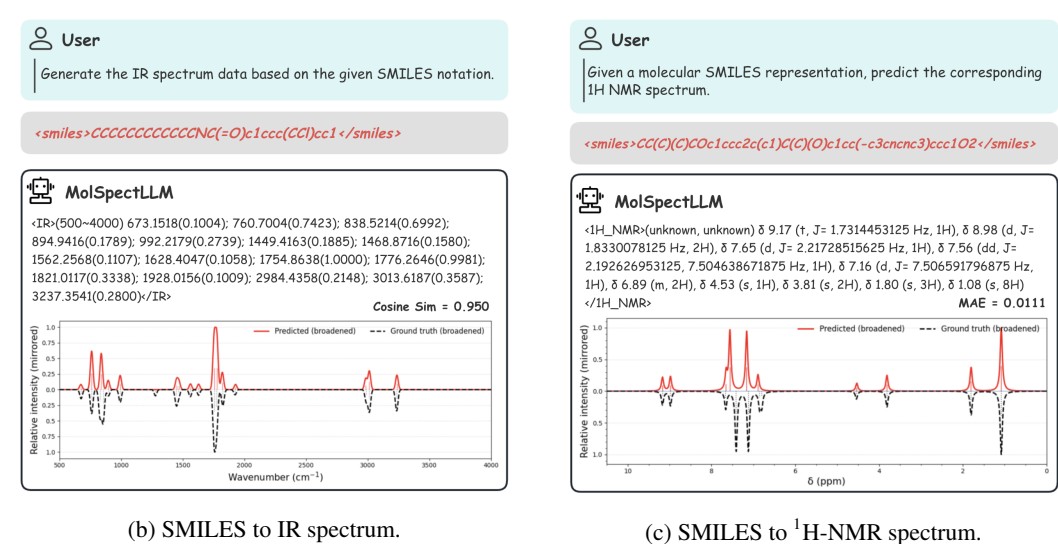

(b) SMILES to IR spectrum.  (c) SMILES to [1]H-NMR spectrum.

Figure 10: Case studies of SMILES-to-Spectra across three different spectra: [1]H-NMR (top), IR (bottom left), and MS (bottom right).

Quantitative results are summarized in Figure 11, which compares MolSpectLLM with several state-of-the-art large language models across four spectrum types ([13]C-NMR, [1]H-NMR, IR, and MS). MolSpectLLM achieves the best performance in all settings, as measured by F1, Jaccard, or cosine similarity, substantially outperforming general-purpose LLMs. Notably, improvements are especially pronounced in NMR spectra, where the model achieves nearly double the F1 score compared with the strongest baseline. These findings demonstrate that MolSpectLLM not only interprets spec-

tra but also generates realistic spectral patterns, bridging structural input and spectroscopic output in a unified modeling framework.

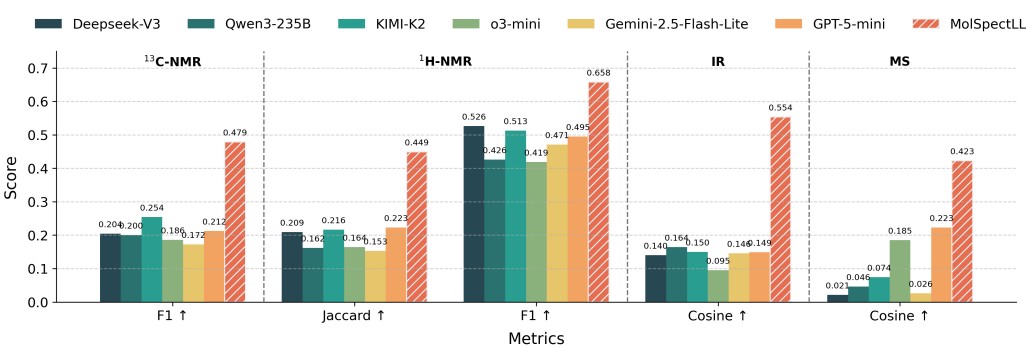

Figure 11: Results on *SMILES-to-Spectra* prediction across four spectrum types with similarity metrics.

### A.4.3 3D STRUCTURE GENERATION

We further evaluate MolSpectLLM on the challenging task of 3D structure generation, where the model is required to predict atomic coordinates, atom types, and bond connectivity directly from symbolic inputs. Figure 12 shows representative case studies for both SMILES-to-3D and IUPAC-to-3D tasks. In each case, the generated structures closely match the ground-truth conformations, demonstrating that MolSpectLLM can reliably capture stereochemistry and spatial constraints from textual molecular representations.

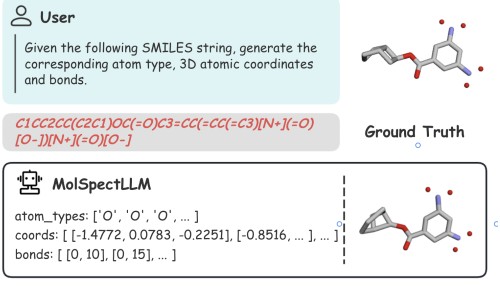

(a) Case study of SMILES to 3D coordinates.

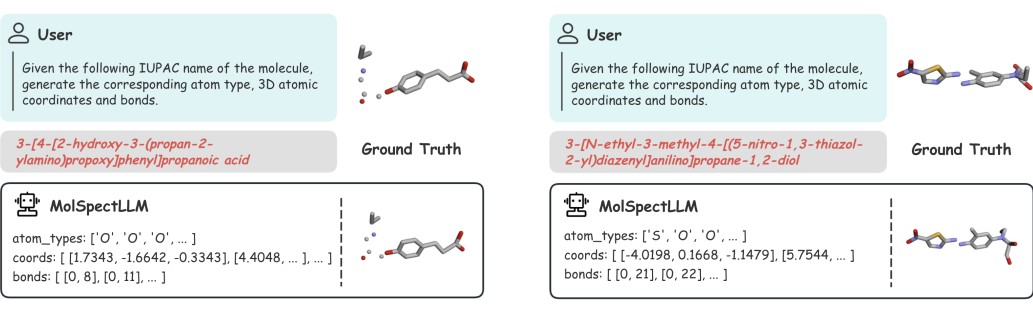

(b) Case study of IUPAC to 3D coordinates (example 1).

(c) Case study of IUPAC to 3D coordinates (example 2).

Figure 12: Case studies of 3D structure generation: SMILES to 3D (top), and IUPAC to 3D (bottom left & right).

Table 4 provides a quantitative comparison against state-of-the-art baselines. On the SMILES-to-3D task, MolSpectLLM achieves the highest structural validity (89.68%), while also maintaining substantially fewer atom clashes and bond violations than most large-scale LLMs. It also reaches the highest fingerprint similarity score (0.582), indicating strong topological agreement with the reference molecules. For the IUPAC-to-3D task, MolSpectLLM again leads in validity (82.78%) and delivers competitive geometry quality, with markedly fewer unrealistic artifacts compared to strong baselines.

These results highlight the unique ability of MolSpectLLM to bridge symbolic notations and geometric molecular space. By accurately generating chemically valid and structurally faithful 3D conformations, MolSpectLLM extends beyond text-only modeling and provides a unified framework that links molecular language with spatial representation, enabling downstream applications in structure-based drug design and molecular property prediction.

Table 4: Results on SMILES-to-3D and IUPAC-to-3D with structural validity and similarity metrics. On each task, the best model is **bolded**.

| Model | SMILES-to-3D | | | | IUPAC-to-3D | | | |
|---|---|---|---|---|---|---|---|---|
| | SDF Valid ($\uparrow$) | Atom Clash ($\downarrow$) | Bond Violation ($\downarrow$) | FP Sim ($\uparrow$) | SDF Valid ($\uparrow$) | Atom Clash ($\downarrow$) | Bond Violation ($\downarrow$) | FP Sim ($\uparrow$) |
| Deepseek-V3 (685B) | 16.50 | 8.941 | 0.151 | 0.152 | 42.50 | **1.138** | 3.543 | **0.721** |
| KIMI-K2 | 22.00 | **0** | **0** | 0.315 | 11.50 | 1.375 | 4.750 | 0.574 |
| o3 | 45.50 | 1.825 | 2.175 | 0.356 | 54.50 | 5.027 | 4.186 | 0.642 |
| Gemini-2.5-Flash | 64.00 | 5.672 | 2.270 | 0.304 | 63.50 | 52.19 | 3.810 | 0.693 |
| GPT-5 | 69.50 | 0.224 | 0.217 | 0.314 | 53.00 | 4.686 | 2.059 | 0.813 |
| MolSpectLLM (7B) | **89.68** | 2.880 | 0.994 | **0.582** | **82.78** | 3.012 | **1.357** | 0.705 |

## A.5 LIMITATIONS

Despite the strong empirical results, several limitations remain. First, we observed that full-parameter fine-tuning can degrade the model's instruction-following ability. This effect likely arises because spectrum-related supervision signals dominate the optimization, overwriting alignment behaviors that were learned during the base model's pretraining. Although our instruction-following SFT stage mitigates this issue to some extent, a residual gap persists, and certain evaluations are still negatively affected.

Second, while MolSpectLLM excels in spectrum-centered and chemistry-specific tasks, its performance on general-purpose tasks and open-domain dialogue is limited compared to much larger language models. This discrepancy may stem from two factors: (i) the comparatively smaller scale of our model and training data, which constrains its ability to generalize beyond chemistry; and (ii) the specialized nature of our fine-tuning, which prioritizes molecular reasoning at the expense of broad coverage.

Together, these observations suggest that future work should explore more balanced adaptation strategies, larger-scale pretraining, and hybrid alignment methods to better preserve instruction-following capability while maintaining strong domain expertise.

## B USAGE OF LANGUAGE MODELS

We use large language model (LLM) to aid in the preparation of this manuscript. Its use was limited to editorial tasks, including proofreading for typographical errors, correcting grammar, and improving the clarity and readability of the text.

