# OpenReview forum: "MolSpectLLM: A Molecular Foundation Model Bridging Spectroscopy, Molecule Elucidation, and 3D Structure Generation"
_ICLR.cc/2026/Conference — ICLR 2026 Conference Withdrawn Submission_

### Official Review · Reviewer_nidD · 2025-10-25

**Soundness:** 1
**Presentation:** 2
**Contribution:** 2
**Rating:** 2
**Confidence:** 4

**Summary:**

The authors present MolSpectLLM, a new LLM finetuned from Qwen2.5-7B for a variety of molecule and spectra-related tasks such as spectra generation, spectra-to-SMILES structural elucidation, and 3D molecule generation. MolSpectLLM achieves improved performance over baseline LLMs on most tasks.

**Strengths:**

MolSpectLLM achieves impressive performance gains over baseline LLMs on several important molecule tasks.

**Weaknesses:**

#### Weaknesses:
- W1: MolSpectLLM does not significantly outperform (and actually underperforms) baseline LLMs on MolQA, which is surprising given that it is finetuned on molecule-related tasks.
- W2: Sequence/token accuracy are not good metrics for SMILES generation. SMILES is not a unique representation for a molecule and it makes more sense to check for Inchi key matches.
- W3: The authors do not provide any information on where the test set comes from for each task, making it difficult for me to contextualize their claims. Also, given the extensive SFT datasets, there may be data leakage in the test set.
- W4: I have some concerns about the SMILES-to-IUPAC conversion. [1] recently found that o3 is capable of doing SMILES-to-IUPAC conversions, yet the authors reports only 2% accuracy. Can the authors share additional details about how these experiments are set up?
- W5: The authors claim they provide standardized representations for spectra as one of their main contributions, but it seems to me these representations are trivial extensions of existing commonly used spectra representations.

[1] Assessing the Chemical Intelligence of Large Language Models, Runcie et al. (2025), https://arxiv.org/abs/2505.07735

**Questions:**

#### Questions:
- Q1: Can the authors include results for the base Qwen2.5-7B for each task? This would be a useful comparison to see how much the proposed SFT improves performance.
- Q2: Where does each test set come from? How were these sets chosen? Have the authors checked that the molecules in the test set are disjoint from the finetuning datasets?
- Q3: Can the authors provide additional detail on how the Spectra-to-SMILES sequence accuracy was calculated? Can the authors also provide the Inchi-key match accuracy?

---

### Official Review · Reviewer_P2cu · 2025-10-29

**Soundness:** 2
**Presentation:** 3
**Contribution:** 3
**Rating:** 6
**Confidence:** 4

**Summary:**

The authors present MolSpectLLM, a molecular LLM that unifies chemistry text and experimental spectroscopy (1H/13C NMR, IR, Raman, UV, MS) with 3D structures. It addresses the gap left by SMILES/text-only models by converting each spectrum type into compact textual descriptors so a single LLM can consume heterogeneous signals. Training proceeds via large-scale pretraining on molecular corpora and spectral data, followed by mixed multi-task SFT (translation between Spectrum/SMILES, SMILES/IUPAC, QA, property prediction, and 3D coordinates) and instruction tuning with lightweight adapters. Reported results include Spectra-to-SMILES sequence accuracy of 15.5%, token accuracy of \~41.7%, and 3D validity around 83\~90%, demonstrating end-to-end multimodal molecular reasoning and generation.

**Strengths:**

- By presenting appropriate text-like conversion methods for diverse spectroscopic data (NMR, IR, MS), the authors successfully integrate these modalities with a pre-trained LLM.

- The trained model exhibits strong performance on a wide range of tasks that involve spectroscopic signals and 3D structural information.

- The accompanying scripts provide rich, reproducible details on model construction, including data preprocessing, training dynamics, and other implementation specifics.

**Weaknesses:**

- Most of the tasks in the experimental results are those the model was directly trained to perform during pretraining for modality alignment. Whether this 3D information confers transferable benefits in more practical fine-tuning scenarios (e.g., biochemical property prediction like MoleculeNet) is not fully explored.

- All quantitative comparisons are made against LLMs pre-trained for the general domain. While this may be inevitable for spectroscopy-related tasks, evaluations such as molecular QA should also include head-to-head comparisons with recent available chemical LLMs [1][2] under identical test conditions.
---
[1] LLaMo: Large Language Model-based Molecular Graph Assistant, NeurIPS 2024

[2] Mol-LLaMA: Towards General Understanding of Molecules in Large Molecular Language Model, NeurIPS 2025

**Questions:**

- It seems like the IUPAC/SMILES conversion task can be done without explicit 3D information; is it necessary to include this task in the pretraining phase? Can the strong performance on this name-conversion task be credibly attributed to the integration of spectroscopic information?

---

### Official Review · Reviewer_3TZJ · 2025-10-30

**Soundness:** 2
**Presentation:** 4
**Contribution:** 2
**Rating:** 2
**Confidence:** 5

**Summary:**

This paper proposes MolSpectLLM, a multi-modal large language model based on Qwen2.5-7B, designed to integrate molecular spectra (NMR, IR, MS), SMILES/IUPAC representations, and 3D molecular structure generation. The proposed method can work in a wide range of different kinds of tasks and bridges spectral analysis, molecular elucidation, and molecular design.

**Strengths:**

1.The approach works in a wide range of tasks, including spectra-to-structure, structure-to-spectra, and 3D structure generation, highlighting its multi-functionality.

2.The paper presents a complete engineering pipeline, with a clear implementation of multi-task training and spectra-to-text transformation.

3.The writing is clear and figures are well-designed, facilitating understanding.

**Weaknesses:**

1.Insufficient experimental rigor and invalid baselines: This is the most critical weakness. The paper claims SOTA performance by comparing their specialized, 7B fine-tuned model against general-purpose, zero-shot Large Language Models. The paper's baselines include DeepSeek-V3 (685B), Qwen3-235B, Kimi-K2, o3, Gemini-2.5-Flash, and GPT-5. These are general-purpose models that have not been trained on the specific, non-natural textual formats for spectra and 3D coordinates that MolSpectLLM was explicitly fine-tuned on.

2.Limited novelty: The concept of spectra textualization is already established, as demonstrated by previous work such as SpectraLLM (https://arxiv.org/abs/2508.08441). While MolSpectLLM extends this framework to additional spectra types and introduces 3D generation, these improvements mainly involve engineering unification rather than novel technical contributions. Characterizing the model as a “foundation model bridging” over claim its methodological innovation, as the principal contribution lies in data pipeline design rather than modeling advances.

3.Misleading claims regarding experimental data: The training datasets are predominantly simulated, including QM9S and the Multimodal Spectroscopic Dataset, while real experimental data is very limited and mostly comes from NMRBank. Claims that the model can handle experimental spectra are not fully substantiated because noise, artifacts, and other real-world experimental variations are largely absent in the training or evaluation.

4.Low absolute performance on core tasks: On the spectra to SMILES task, the model achieves only 15.50% sequence accuracy, indicating failure in approximately 85% of cases. This low performance suggests that the model is far from being practically useful or reaching state-of-the-art standards.

**Questions:**

please respond to the weakness

---

### Official Review · Reviewer_WfzJ · 2025-10-31

**Soundness:** 2
**Presentation:** 2
**Contribution:** 2
**Rating:** 2
**Confidence:** 3

**Summary:**

The paper proposes MolSpectLLM, a language-based molecular foundation model that operates on molecular spectra data. It introduces a new textual format to represent spectra, arguing that this overcomes sparsity in raw spectral vectors and enables an LLM to perform a broad set of tasks: name conversions, molecular QA, and 3D from SMILES.

The main claims are:
1. Most existing molecular foundation models rely on string-based representations, discarding 3D structure and experimental molecular spectra (line 45-50). Incorporating molecular spectroscopy and 3D structure into a unified molecular foundation model yields strong performance (line 67-68).
2. Directly feeding raw spectral vectors into LLMs is ineffective because spectra data are high-dimensional yet extremely sparse (line 141-142). The proposed textual formats for spectral data efficiently encode spectral information and enables LLMs to effectively interpret and leverage experimental data (line 69-71).
3. New evaluation metrics provides a more direct, interpretable, and efficient means of assessing spectral fidelity (Line 72-74)

**Strengths:**

* Bringing experimental spectroscopy into foundation models is important, and a unified interface that spans spectra↔structure tasks is valuable in principle.
* The proposed method seems to work over multiple evaluation tasks.

**Weaknesses:**

* MolSpectLLM is explicitly pre-trained on chemistry corpora and then supervised-fine-tuned for the exact evaluation tasks (QA for 3D generation, spectral interpretation, motif identificaation, name conversion, spectrum generation). However, the quantitative comparisons are reported only against general-purpose LLMs, not against chemistry-fine-tuned LLMs. It is expected that a chemistry pretrained and fine-tuned LLMs would perform better on chemistry tasks compared to general-purpose LLMs, weakening claim 1.
* The paper positions MolSpectLLM as a molecular foundation model but does not compare to molecule-native foundation models, which some of them are also 3D-aware (UniMol, GraphMVP) or Spectra-aware (Mol-Spectra).
* The paper claims that directly feeding raw spectral vectors into LLMs is ineffective, but there is no ablation to verify this claim.
* The paper reports IUPAC ↔ SMILES results against general LLMs, but did not compare to classical cheminformatics tools such as OPSIN. Also, it is unclear why name conversion is an important evaluation task for a molecular foundation model.

Minor:
* Some paragraphs have confusing chronology. For example, line 100-102 cited works from 2024-2025 as "early works" that lack 3D structural information. Then, line 105-106 introduces works from 2021-2022 as recent advances that attempt to address the gap. This also contradicts Claim 1 that existing models rely mostly on string-based representation.
* Technical terms are not explained sufficiently. For example, (line 147) 13C NMR and 1H NMR are explained only in the Appendix, but there is no direct reference to them in the main text. (line 157) Standardizing spectra is claimed to removes sparsity and noise, but no explanation on what kind of noise and how the standardization removes noise. The molecular QA task is not well-described in the submission.
* Figure 3 caption: Exmaple —> Example

**Questions:**

Please refer to weaknesses.

---

### Note · Authors · 2025-11-20

I have read and agree with the venue's withdrawal policy on behalf of myself and my co-authors.